# *SNCA* Gene Methylation in Parkinson's Disease and Multiple System Atrophy

Ekaterina Yu. Fedotova *, Elena V. Iakovenko, Natalia Yu. Abramycheva and Sergey N. Illarioshkin

Research Center of Neurology, 125367 Moscow, Russia
* Correspondence: fedotova@neurology.ru

**Abstract:** In recent years, epigenetic mechanisms have been implicated in the development of multifactorial diseases including neurodegenerative disorders. In Parkinson's disease (PD), as a synucleinopathy, most studies focused on DNA methylation of *SNCA* gene coding alpha-synuclein but obtained results were rather contradictory. In another neurodegenerative synucleinopathy, multiple system atrophy (MSA), very few studies investigated the epigenetic regulation. This study included patients with PD (n = 82), patients with MSA (n = 24), and a control group (n = 50). In three groups, methylation levels of CpG and non-CpG sites in regulatory regions of the *SNCA* gene were analyzed. We revealed hypomethylation of CpG sites in the *SNCA* intron 1 in PD and hypermethylation of predominantly non-CpG sites in the *SNCA* promoter region in MSA. In PD patients, hypomethylation in the intron 1 was associated with earlier age at the disease onset. In MSA patients, hypermethylation in the promotor was associated with shorter disease duration (before examination). These results showed different patterns of the epigenetic regulation in two synucleinopathies—PD and MSA.

**Keywords:** Parkinson's disease; multiple-system atrophy; DNA methylation; *SNCA* gene





## 1. Introduction

The hallmark of all neurodegenerative diseases is the accumulation of pathological aggregates of various proteins in the neural and/or glial cells. Therefore, these diseases can also be referred to as proteinopathies [1]. In synucleinopathies, such as Parkinson's disease (PD) and multiple system atrophy (MSA), pathological aggregates of the small synaptic protein alpha-synuclein are found in the central and peripheral nervous system [2].

PD is the second most common neurodegenerative disease in the world after Alzheimer's disease [3–6]. The clinical picture of PD includes motor symptoms such as bradykinesia, rigidity, rest tremor, and postural instability and a variety of non-motor symptoms (autonomic, cognitive, affective, etc.) [7]. The main pathomorphological characteristic of PD is a loss of dopaminergic neurons in the substantia nigra pars compacta, with the accumulation of pathological aggregates of alpha-synuclein in Lewy bodies [8].

PD is a multifactorial disease with a documented substantial increase in its prevalence in the past decades; both environmental and genetic factors contribute to the pathology of this disorder [9]. To date, more than 20 genes and loci have been associated with hereditary forms of PD [10]. In general, genetic forms account for about 10–20% of all cases of PD [11]. The main genes involved in PD are *SNCA*, *PARK2* (*Parkin*), *LRRK2*, *DJ-1*, *PINK1*, and *GBA* [12]. Furthermore, genome-wide association studies (GWAS) revealed many polymorphisms in PD-associated genes, including the *SNCA* gene that encodes alpha-synuclein [13–15]. However, despite such a wide genetic palette, there is a phenomenon of "missing heredity" in PD, which can be partly explained by an influence of epigenetic modifications [16].

MSA is an incurable, progressive neurodegenerative disease with a late onset also belonging to the group of synucleinopathies. The clinical picture may include parkinsonism with poor response to levodopa therapy, cerebellar ataxia, autonomic dysfunction, and pyramidal signs [17]. A pathological feature of MSA is the presence of oligodendroglial cytoplasmic inclusions containing aggregated alpha-synuclein as well as, to a lesser extent, neuronal cytoplasmic inclusions of alpha-synuclein aggregates, which distinguishes this condition from PD [18]. Oligodendroglial inclusions can occur in various areas of the brain, which determines the clinical picture of MSA. Striatonigral degeneration is more pronounced in patients with a parkinsonian phenotype, i.e., MSA-P, and olivopontocerebellar degeneration is more pronounced in patients with a cerebellar phenotype, i.e., MSA-C [19].

MSA is also considered a multifactorial disease, and several nucleotide polymorphisms in *SNCA*, *MAPT*, and *COQ2* genes have been shown to represent possible risk factors for the development of MSA [20–22]. However, in a recent large MSA GWAS study, no loci with a strong association with the disease were found [23].

In recent years, epigenetic mechanisms have been implicated in the development of multifactorial diseases (mainly studied in cancer). There is also an increase in the number of publications on epigenetics in neurodegenerative diseases, including PD [24]. Among all epigenetic modifications, DNA methylation is the most frequently studied mechanism in PD. Most DNA methylation studies in PD have focused on the *SNCA* gene. Alpha-synuclein is a protein consisting of 140 amino acids localized in presynaptic terminals and involved in the regulation of synaptic vesicles [25]. The *SNCA* gene encoding alpha-synuclein consists of six exons, where exon 1 is non-coding. Transcriptionally significant regions are localized in intron 1. At the 5'-end of the *SNCA* gene, in the regulatory region containing promoter, non-coding exon 1, and intron 1, there is a 591 bp CpG island. Several studies have been performed on the methylation of this region, but the data obtained are rather contradictory. In most studies, hypomethylation of CpG sites in the region of *SNCA* intron 1 was found [26–31], but there are also opposite data, where differences between groups of patients with PD and controls could not be identified [32–34]. Currently, there are very few studies on an influence of epigenetic regulation on the development of MSA [35].

## 2. Results

PD and control groups did not differ in the methylation levels of CpG and non-CpG sites in the promoter region of the *SNCA* gene. At the same time, statistically significant differences were found in the methylation levels of four CpG sites in intron 1 (Table 1, Figure 1). These CpG sites were significantly hypomethylated in the PD group compared to controls.

**Table 1.** Methylation levels of CpG sites in intron 1 in patients with PD and in control group.

| CpG | PD | C | $p$(U) * |
|---|---|---|---|
| 31 | 0 [0; 10] | 10 [0; 17] | 0.000221 |
| 32 | 0 [0; 12.25] | 14 [0; 19] | 0.000627 |
| 33 | 0 [0; 9.5] | 10 [0; 15] | 0.000683 |
| 45 | 40 [35; 48] | 50 [40; 55] | 0.000935 |

* $p < 0.0017$. PD, Parkinson's disease; C, control group.

Differences in intron 2 were also revealed. In PD patients, significant hypermethylation of CpG-2 was shown in comparison to controls: 100 [87; 100]% vs. 83 [81.5; 100]%, respectively, $p$(U) = 0.000007 (Figure 2).

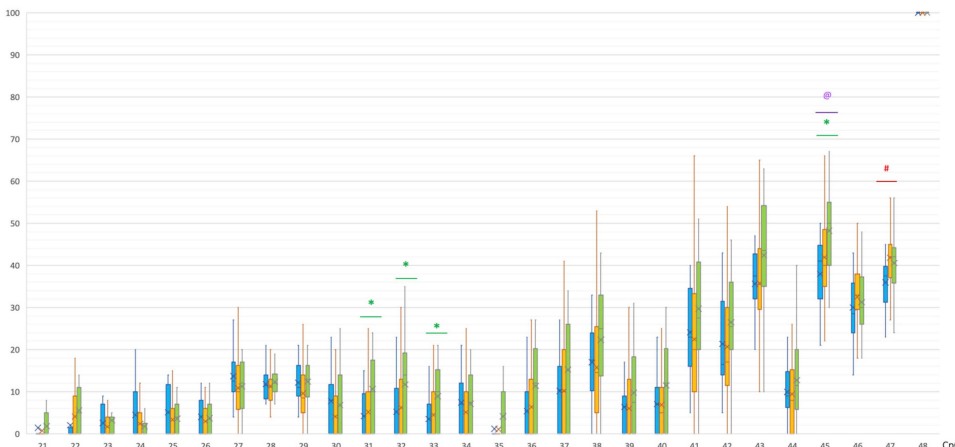

**Figure 1.** Methylation of CpG sites in intron 1 of the *SNCA* gene. @, statistically significant differences between MSA and control groups; *, statistically significant differences between PD and control groups; #, statistically significant differences between MSA and PD groups. Blue, MSA group; yellow, PD group; green, control group.

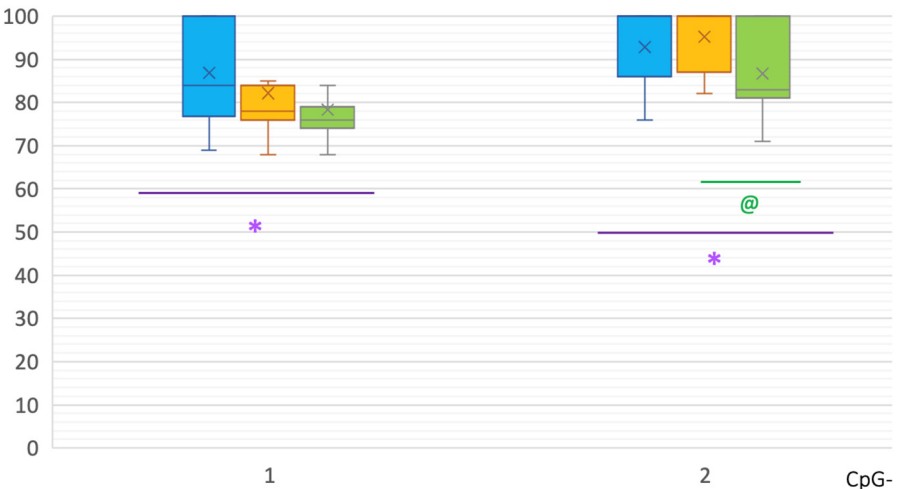

**Figure 2.** Methylation of CpG sites in intron 2 of the *SNCA* gene. *, statistically significant differences between MSA and control groups; @, statistically significant differences between PD and control groups. Blue, MSA group; yellow, PD group; green—control group.

The MSA group was characterized by a different methylation pattern. Statistically significant differences between MSA and controls were found in all studied regions. In intron 1, a CpG-45 site was hypomethylated in the MSA group compared to the control group: 41 [32; 44.25]% vs. 50 [40;55]%, *p*(U) = 0.000183 (Figure 1). Moreover, in the MSA group, one site CpG-47 in intron 1 was less methylated than in PD group: 37.5 [31.75; 39.25]% vs. 42 [37; 45]%, *p*(U) = 0.000228.

In intron 2 of the *SNCA* gene in MSA patients, two CpG sites showed hypermethylation compared to controls: CpG-1 84 [77.75; 100]% vs. 77 [74; 79]%, *p*(U) = 0.001268; and CpG-2 100 [86; 100]% vs. 83 [81.5; 100]%, *p*(U) = 0.008788 (Figure 2). No differences were found between MSA and PD patients in the methylation levels in intron 2.

In the promoter, a CpG-7 site was significantly hypermethylated in the MSA group compared to the control group (36 [31; 41.5]% vs. 26 [21; 33]%, *p*(U) = 0.001076, Figure 3) and PD group (29 [23.5; 31]%, *p*(U)= 0.000240).

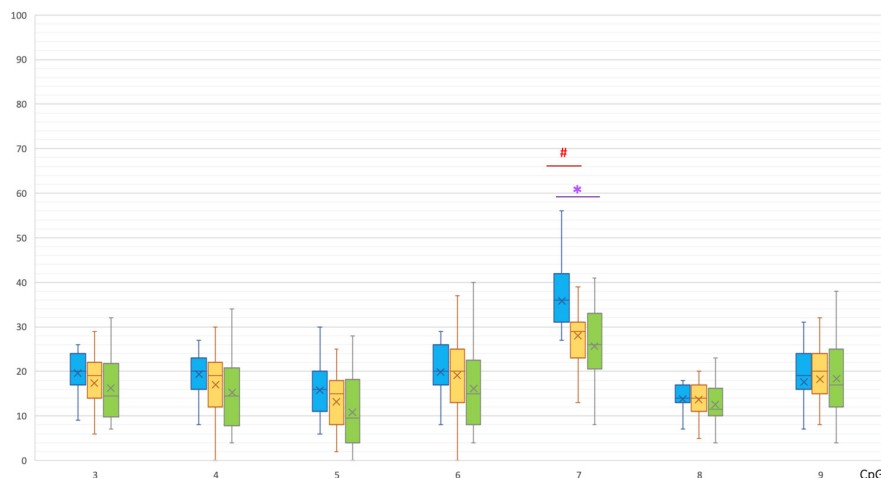

**Figure 3.** Methylation of CpG sites in the promoter region of the *SNCA* gene. *, statistically significant differences between MSA and control groups; #, statistically significant differences between MSA and PD groups. Blue, MSA group; yellow, PD group; green, control group.

In the MSA group, statistically significant hypermethylation of eight non-CpG sites of the *SNCA* promoter region was detected in comparison with the control group (Table 2, Figure 4).

**Table 2.** Methylation levels of non-CpG sites in the *SNCA* promoter in MSA patients and controls.

| Non-CpG | MSA | C | *p*(U) * |
|---|---|---|---|
| 3A | 13 [9.5; 14] | 6 [3.25; 8] | 0.000055 |
| 4B | 13 [11; 17.5] | 7 [4.25; 10] | 0.000730 |
| 4C | 23 [19.5; 28] | 12 [7; 19] | 0.000898 |
| 4I | 26 [23; 31] | 13 [9.5; 21.25] | 0.000150 |
| 4J | 28 [21.5; 30.5] | 14 [10.75; 21.5] | 0.000180 |
| 4K | 27 [25; 32] | 19 [12.75; 26] | 0.001016 |
| 4L | 10 [8; 13] | 6 [3.25; 9] | 0.000936 |
| 7A | 39 [32; 43] | 13 [9; 19] | 0.0000001 |

* $p < 0.0017$. MSA, multiple-system atrophy; C, control group.

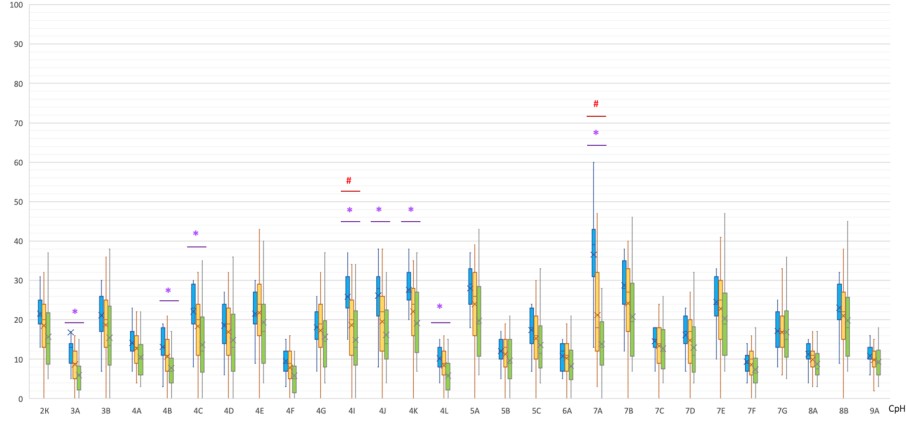

**Figure 4.** Methylation of non-CpG sites in the promoter region of the *SNCA* gene. *, statistically significant differences between MSA and control groups; #, statistically significant differences between MSA and PD groups. Blue, MSA group; yellow, PD group; green, control group.

A statistically significant difference was also found in the methylation levels of the two non-CpG sites between the MSA and PD groups (both were hypermethylated in MSA): (1) CpG-4I, 26 [23; 31]% vs. 20 [11.5; 24.5]%, *p*(U) = 0.000947; (2) CpG-7A, 39 [32; 43]% vs. 18 [12; 32]%, *p*(U) = 0.000059 (Figure 4).

As can be seen in Figures 1–4, PD patients had an intermediate methylation level that fell between that of MSA patients and the control group in all examined *SNCA* regulatory regions. We revealed hypomethylation of *SNCA* intron 1 in PD and hypermethylation of both CpG and non-CpG sites of the *SNCA* promoter region in MSA. In both PD and MSA, intron 2 was hypermethylated in the "body" of the *SNCA* gene.

We analyzed the association of methylation with the clinical and demographic features. In the PD, MSA, and control groups, no statistically significant differences were found between men and women in methylation levels of the studied *SNCA* regions.

In the PD group, regression analysis was performed to determine the effect of age and disease onset on methylation levels in *SNCA* regions. No correlations were found for sites in the promoter and intron 2. Statistically significant direct correlations were found between the age of onset and methylation levels of four CpG sites in *SNCA* intron 1: CpG-22 (r = 0.52), CpG-26 (r = 0.520, CpG-39 (r = 0.52), and CpG-40 (r = 0.56) (for all, *p* = 0.000001). The results are shown in Figure 5. Thus, decreased methylation levels in intron 1 are associated with earlier disease onset of PD.

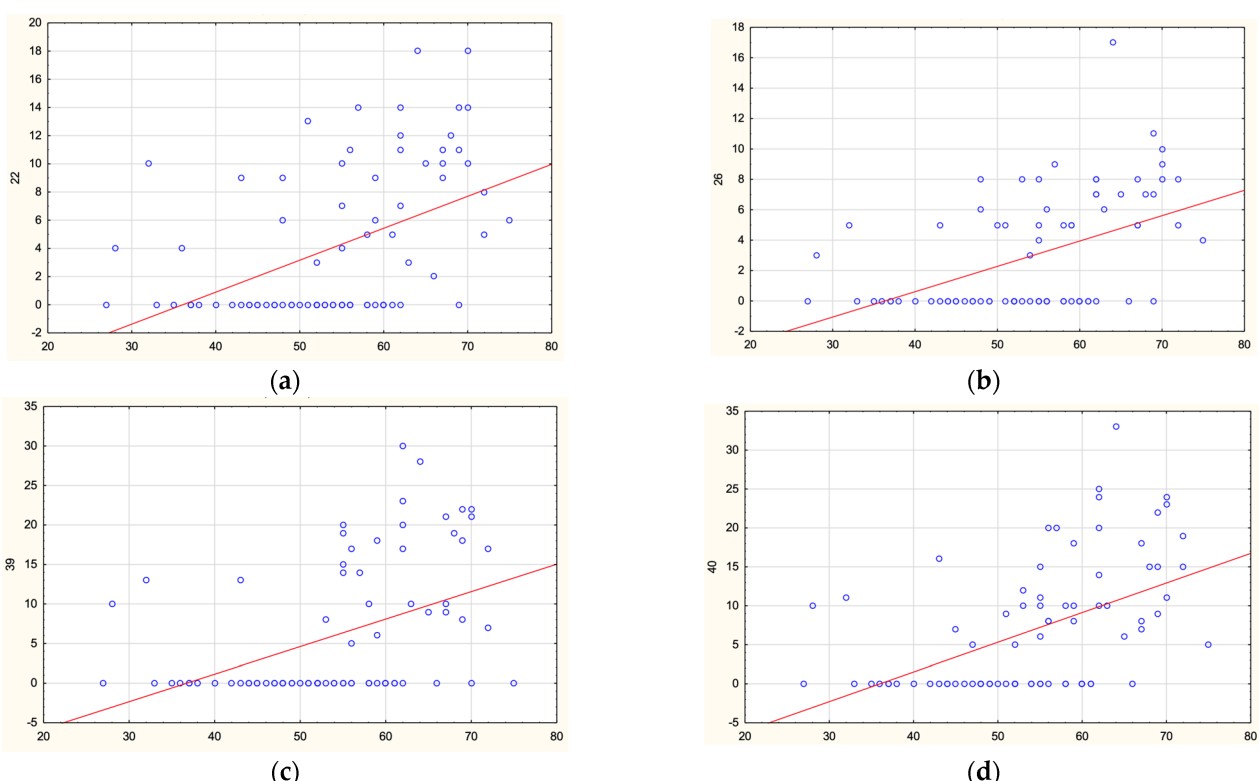

**Figure 5.** Correlations between age of PD onset and methylation levels of CpG-22 (**a**), CpG-26 (**b**), CpG-39 (**c**), and CpG-40 (**d**) in *SNCA* intron 1.

In patients with MSA, correlation with age at the disease onset was not found.

We studied correlations between methylation levels and disease duration (before examination). In the PD group, no statistically significant correlations were found. In the MSA group, a strong inverse correlation was found between the disease duration and the level of methylation for three non-CpG sites in the promoter region of the *SNCA* gene: non-CpG-4I (r = −0.66, *p* = 0.00066), non-CpG-4J (r = −0.63, *p* = 0.00131), and non-CpG-4K (r = −0.67, *p* = 0.00043). The results are shown in Figure 6. Thus, the higher methylation levels in these identified non-CpG sites were associated with the shorter disease duration of MSA. In the other regulatory regions of *SNCA*, no correlations were found.

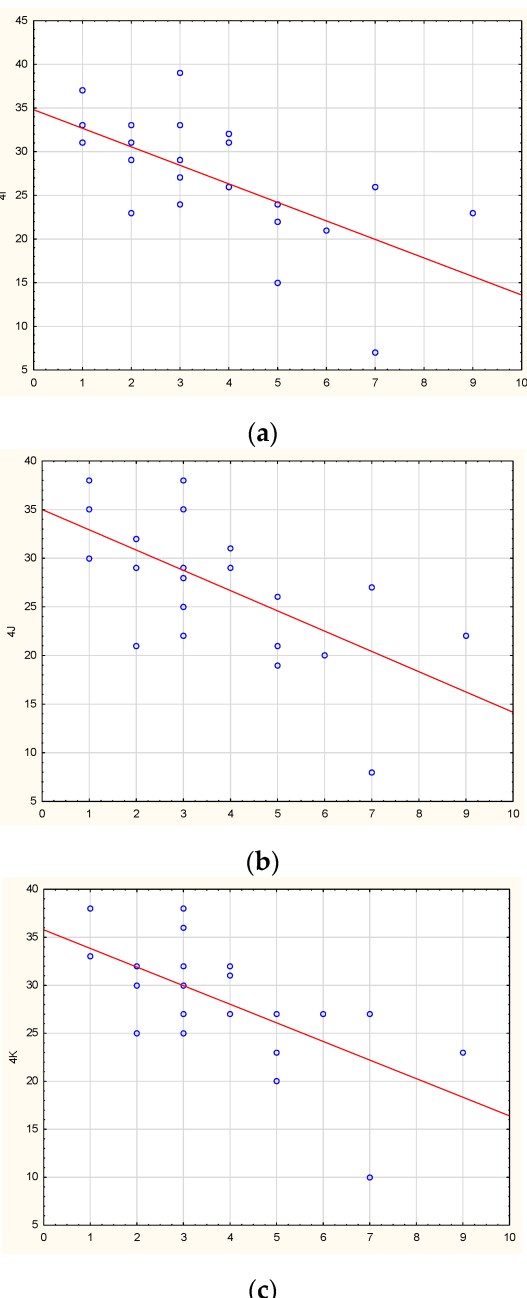

**Figure 6.** Correlations between the disease duration in MSA patients and methylation levels of non-CpG-4I (**a**), non-CpG-4J (**b**), non-CpG-4K (**c**) in the *SNCA* promoter.

We analyzed associations between *SNCA* methylation levels and disease severity according to the Hoehn–Yahr scale and to the Unified Parkinson's Disease Rating Scale—part III (UPDRS-III) and found no correlations.

Treatment with any antiparkinsonian drugs did not affect the methylation profile of the *SNCA* gene. Treatment by levodopa and amantadine did not change the methylation pattern. There was only a tendency of a difference between subgroups of patients taking and not taking dopamine receptor agonists: hypomethylation of CpG-38 in intron 1 was found in the agonist subgroup (10 [1; 19.25]% vs. 20 [10; 30]% in patients without agonists, $p = 0.0028$), but this difference became not statistically significant after Bonferroni correction.

Thus, hypomethylation in intron 1 was associated with earlier age at the disease onset in PD patients. In MSA patients, hypermethylation in the promotor region was

associated with shorter disease duration and hence possibly with more rapid progression of the disease. No other clinical–epigenetic correlations were found.

## 3. Discussion

The first studies on methylation patterns in PD focused on their assessment of epigenetic changes in pathomorphological samples of different brain areas. Jowaed et al. [26] and Matsumoto et al. [27] showed significant hypomethylation of CpG sites in *SNCA* intron 1 in the substantia nigra in PD patients. Jowaed showed hypomethylation of CpG-8, 12, and 17 (designated CpG-28, 32, and 37 in the current study). Further works assessed the levels of methylation in peripheral blood of patients with PD, and the results of these works were contradictory; in most studies, hypomethylation in *SNCA* intron 1 was detected when compared with healthy volunteers [28–31]. In the studies of Richter et al. [32] and Song et al. [33], no differences were found, which can be explained by small samples of patients. Tan et al. revealed hypomethylation of CpG-31, 33, and 38 in *SNCA* intron 1 (according to the nomenclature in our study) [29]. Ai et al. showed hypomethylation of CpG-26, 28, and 29 in the same region [28]. Schmitt et al. studied 490 PD patients and found statistically significant hypomethylation in PD in all studied 14 CpG sites in *SNCA* intron 1 (CpG 22–27 and 36–43) [31]. Interestingly, one study showed that in PD cases the DNA methylation pattern in brain tissue resembled that in peripheral blood [36].

In our study, significant differences were found between patients with PD and healthy volunteers in the level of methylation of four CpG sites (out of 28) in intron 1 of the *SNCA* gene, namely CpG-31, 32, and 33 (CpG-11, 12, and 13, according to Jowaed et al.), and the CpG-45 sites were significantly hypomethylated in patients with PD. It is known that exon 1 of *SNCA* is non-transcribed; therefore, the regulatory region of the *SNCA* gene responsible for transcription processes is located before exon 2 in the region of intron 1. A number of studies determined binding sites for various transcription factors there [26,37,38]. It is noteworthy that a conservative binding site for a transcription factor of GATA family, which is involved in the regulation of *SNCA* expression, is located within the region of *SNCA* intron 1, which includes CpG sites 22 to 43. As a number of key transcription factors attach to this region, the hypomethylation at CpG-31, 32, and 33 that we found may be directly related to an increased expression of alpha-synuclein observed in synucleinopathies [37,39,40].

In MSA, we found the differently methylated *SNCA* promoter region, eight non-CpG sites, and one CpG were statistically significantly hypermethylated, wherein the only differently methylated CpG site (CpG-7, chr4: 89,837,952) was located near another non-CpG site (CpG-7A, chr4: 89,837,949). Non-CpG methylation is more prevalent in cells of the nervous system compared to other cells in the body [41]. The functions of non-CpG methylation are still poorly understood, and it is assumed that they are also associated with the processes of expression and tissue specificity [42]. Currently, due to a lack of studies on non-CpG site methylation in neurodegenerative diseases, we were unable to compare our results with others.

In PD and MSA, we also revealed significant hypermethylation in intron 2 of the *SNCA* gene. A high level of methylation within a gene could serve as an indicator of an active gene transcription and is considered a fairly common phenomenon. On the other hand, hyper- and hypomethylation of certain regions within genes can affect alternative splicing, which occurs in the case of alpha-synuclein with its several spliced transcripts [43]. However, further research is needed on this issue.

There are very few works on the DNA methylation in MSA cases. In one epigenome-wide association study (EWAS), in the white matter of MSA patients, 79 differentially methylated regions in the genome were identified, and one of these regions was a CpG island in the *SNCA* gene [35]. Mostly, hypermethylation of CpG sites was detected in MSA in this study, which is consistent with our data on the hypermethylation of CpG and non-CpG sites in the promoter and intron 2 of the *SNCA* gene.

In general, the level of methylation in PD is intermediate between patients with MSA and the control group (Figures 1–4). This corresponds to the clinical picture of PD, which is "less aggressive" and "more benign" compared to MSA cases.

In the present study, a correlation analysis between methylation levels and clinical manifestations in PD and MSA was performed. The work demonstrated direct correlations between the methylation of four CpG sites in *SNCA* intron 1 and the age of disease onset in the PD group. Thus, the lower the methylation level that occurs, the earlier disease develops. This correlation is consistent with the data presented by Schmitt et al. [31]. Thus, the hypomethylation in *SNCA* intron 1 is associated with PD and its early onset.

In the MSA group, three strong inverse correlations were found between the duration of the disease and the level of methylation of non-CpG sites in the *SNCA* promoter region. Hence, the higher the methylation level that occurs, the shorter disease duration until the neurological examination. It should be noted that all three methylated cytosines are located next to each other and are arranged in a row «CCC» (chr4:89,837,992-4), and all these three cytosines are hypermethylated in MSA compared to controls. Thus, the hypermethylation in *SNCA* promoter is associated with MSA and the disease duration.

We were unable to identify correlations between the level of methylation and the disease severity on the Hoehn–Yahr scale and UPDRS-III subscale in PD. In addition, in our study, there was no association between the level of methylation and the presence of levodopa/dopamine receptor agonists/amantadines therapy in PD patients, whereas Schmitt et al. in in vitro experiments demonstrated an elevation of *SNCA* intron 1 methylation in cells exposed to levodopa. Thereby, levodopa may have a protective function, as they supposed [31].

In conclusion, this study showed the involvement of epigenetic regulation in the development of synucleinopathies, the difference in the methylation pattern between neuronal and glial forms of neurodegeneration (PD vs. MSA), and its associations with some clinical features.

## 4. Materials and Methods

We recruited 82 patients with a diagnosis of PD, including 41 women and 41 men (age: 62 [54.5; 67] years). The diagnosis was determined according to the criteria of International Parkinson and Movement Disorder Society [44]. Age at the disease onset was 55 [48; 62] years, and the disease duration was 4 [3; 8] years. The distribution of PD patients by Hoehn–Yahr scale was as follows: stage 1—14 patients (17.2%); stage 2—26 (31.6%); stage 3—40 (48.8%); and stage 4—2 (2.4%). Mean score on UPDRS-III was 45 [22; 56]. The majority of patients (64 patients, 78%) received antiparkinsonian therapy: levodopa therapy—44 (53.7%) patients, dopamine receptor agonists—40 (48.8%) patients, and amantadine—31 (37.8%) patients.

The MSA group included 24 patients with a parkinsonian phenotype (MSA-P): 7 men and 17 women aged 58.5 [55.5; 69] years. The diagnosis was determined according to the criteria of International Parkinson and Movement Disorder Society [17]. In MSA group the age at the disease onset was 55 [52; 64] years, and the disease duration 3.5 [2; 5.25] years.

A control group included 50 neurologically healthy volunteers (31 women, 19 men; age: 58.5 [53; 62.75] years). All three groups were sex- and age-matched.

The location of studied CpG sites in a reference genome (GRCh38) is presented in Appendices A and B. Methylation level was determined in the promoter region (at the 5′-end, CpG 3–9, numbering of CpG sites from the beginning of the promoter region), at the 3′-end of intron 1 (CpG 21–48 of 48, numbering from the beginning of intron 1), and in intron 2 (CpG 1–2 of 2) of the *SNCA* gene. As preliminary experiments showed that the region of the 5′-end of intron 1 of *SNCA* (CpG 1–20) was demethylated both in patients and in controls, we did not further study the methylation level in these CpG sites.

In addition to CpG sites, we also searched for and analyzed non-CpG sites (CpH sites). Non-CpG sites were detected by sequence analysis of methylated cytosine (mC) preceding adenine, thymine, or cytosine (mCA, mCT, and mCC, respectively). In the promoter region of the *SNCA* gene, clustered non-CpG methylation sites (28 non-CpG sites)

were found, which were also analyzed. (Non-CpG sites were marked by a combination of the number of the preceding CpG site and a Latin letter in alphabetical order.) A search was also conducted for non-CpG sites in intron 1 and intron 2 of the *SNCA* gene, but no such non-CpG sites were found in this region in a group of patients and healthy volunteers.

Genomic DNA samples were obtained from peripheral blood leukocytes by Wizard Genomic DNA Purification Kit (Promega, USA) and underwent bisulfite conversion using the EZ DNA Methylation Kit (Zymo Research, USA). Then, the converted gDNA was used as template for PCR amplification. Primers flanking the studied regions were selected using the MethPrimer software, and their synthesis was performed in Syntol company (Moscow). The primer design is shown in Table 3.

**Table 3.** Primers for *SNCA* gene methylation analysis.

| Region | | | Forward Primer | Reverse Primer |
|---|---|---|---|---|
| | promoter | | TAGAAGGGGTTGAAGAAGAAAATTG | AAACTCAACAAATCCTCTTTCCA |
| intron 1 | | (1) | GTTTAAGGAAAGAGATTTGATTTGG | TTACCACCTATTAACTTAACCTC |
| intron 1 | | (2) | GAGGTTAAGTTAATAGGTGGTAA | AAATATCCTTAACATAAATCCCAAAA |
| intron 1 | | (3) | TTTTGGGATTTATGTTAAGGATATTT | ATAACTAATAAATTCCTTTACACCAC |
| | intron 2 | | GTTTGTTAAAAAGGTGGATTGAGT | CTTTATACACATCACAAAAACATATC |

Analysis of the methylation of the *SNCA* gene was carried out by direct Sanger sequencing. Sequences were analyzed on ABI Prism 3130 analyzer (Applied Biosystems) using Data Collection Software v3.0. The level of methylation was defined by analyzing Sanger sequencing results. The percentage of methylation for each CpG site for each DNA sample was calculated as a ratio of a blue peak «C» height (this peak indicates the presence of a methylated cytosine) to a sum of «C+T» blue and red peaks height in this position (methylated blue and unmethylated red cytosine). For non-CpG sites, the calculation was carried out similarly. The assessment was performed using the SeqBase computer program developed at Research Centre of Medical Genetics (Moscow). This program takes into account the non-equivalence of sequenced nucleotide composition as a result of a bisulfite conversion [45].

Statistical analysis was performed in Statistica 13.3 (TIBCO Software Inc.). To compare two independent groups, the Mann–Whitney U-test was used. To compare several independent groups, the Kruskal–Wallis test was used. Correlation analysis was carried out using Spearman's rank test. The correlation was assessed as weak, with r ranged from 0.3 to 0.5 (and was not taken into account in the study), moderate with r = 0.5–0.7, and strong with r = 0.7–0.9. Results were considered statistically significant if $p < 0.05$. A Bonferroni correction was applied due to multiple comparisons. Therefore, statistical significance level for promoter 28 CpG sites and 28 non-CpG sites was 0.007; for 7 CpG sites of intron 1 was 0.0017; and for 2 CpG sites of intron 2 was 0.025.

**Author Contributions:** Conceptualization, E.Y.F.; methodology, N.Y.A.; formal analysis, N.Y.A.; investigation, N.Y.A. and E.V.I.; resources, E.V.I.; writing—original draft preparation, E.V.I. and E.Y.F.; writing—review and editing, E.Y.F. and S.N.I.; visualization, E.V.I.; supervision, S.N.I. All authors have read and agreed to the published version of the manuscript.

**Funding:** This research received no external funding.

**Institutional Review Board Statement:** The study was conducted in accordance with the Declaration of Helsinki and approved by the Local Ethics Committee of Research Center of Neurology (protocol #3-4/19, 27 March 2019).

**Informed Consent Statement:** Informed consent was obtained from all subjects involved in the study.

**Data Availability Statement:** Not applicable.

**Conflicts of Interest:** The authors declare no conflict of interest.

## Appendix A

CpG sites and non-CpG sites in the promoter region of the *SNCA* gene and their location on the genome browser.

| Sites | Nucleotides following the Methylated Cytosine | Location of Cytosine (GRCh38.p13chr4) (NC_000004.12) |
|---|---|---|
| non-CpG-2K | mCCG | chr4: 89,838,054 |
| CpG-3 | mCG | chr4: 89,838,053 |
| non-CpG-3A | mCTA | chr4: 89,838,046 |
| non-CpG-3B | mCAC | chr4: 89,838,040 |
| CpG-4 | mCG | chr4: 89,838,038 |
| non-CpG-4A | mCTG | chr4: 89,838,036 |
| non-CpG-4B | mCTC | chr4: 89,838,026 |
| non-CpG-4C | mCCC | chr4: 89,838,024 |
| non-CpG-4D | mCCA | chr4: 89,838,023 |
| non-CpG-4E | mCAG | chr4: 89,838,022 |
| non-CpG-4F | mCTG | chr4: 89,838,007 |
| non-CpG-4G | mCAA | chr4: 89,838,004 |
| non-CpG-4I | mCCC | chr4: 89,837,994 |
| non-CpG-4J | mCCA | chr4: 89,837,993 |
| non-CpG-4K | mCAT | chr4: 89,837,992 |
| non-CpG-4L | mCTA | chr4: 89,837,987 |
| CpG-5 | mCG | chr4: 89,837,981 |
| non-CpG-5A | mCCT | chr4: 89,837,974 |
| non-CpG-5B | mCTT | chr4: 89,837,973 |
| non-CpG-5C | mCCG | chr4: 89,837,966 |
| CpG-6 | mCG | chr4: 89,837,965 |
| non-CpG-6A | mCTT | chr4: 89,837,963 |
| CpG-7 | mCG | chr4: 89,837,952 |
| non-CpG-7A | mCTG | chr4: 89,837,949 |
| non-CpG-7B | mCCT | chr4: 89,837,933 |
| non-CpG-7C | mCTT | chr4: 89,837,932 |
| non-CpG-7D | mCCA | chr4: 89,837,926 |
| non-CpG-7E | mCAC | chr4: 89,837,925 |
| non-CpG-7F | mCTG | chr4: 89,837,923 |
| non-CpG-7G | mCAG | chr4: 89,837,910 |
| CpG-8 | mCG | chr4: 89,837,907 |
| non-CpG-8A | mCTG | chr4: 89,837,904 |
| non-CpG-8B | mCAG | chr4: 89,837,900 |
| CpG-9 | mCG | chr4: 89,837,896 |
| non-CpG-9A | mCTG | chr4: 89,837,894 |

## Appendix B

CpG sites in the intron 1 and intron 2 of the *SNCA* gene and their location on the genome browser.

| Sites | Location of cytosine (GRCh38.p13chr4) (NC_000004.12) |
| --- | --- |
| intron 1 CpG-21 | chr4: 89,836,561 |
| intron 1 CpG-22 | chr4: 89,836,505 |
| intron 1 CpG-23 | chr4: 89,836,498 |
| intron 1 CpG-24 | chr4: 89,836,488 |
| intron 1 CpG-25 | chr4: 89,836,481 |
| intron 1 CpG-26 | chr4: 89,836,479 |
| intron 1 CpG-27 | chr4: 89,836,461 |
| intron 1 CpG-28 | chr4: 89,836,420 |
| intron 1 CpG-29 | chr4: 89,836,383 |
| intron 1 CpG-30 | chr4: 89,836,329 |
| intron 1 CpG-31 | chr4: 89,836,311 |
| intron 1 CpG-32 | chr4: 89,836,309 |
| intron 1 CpG-33 | chr4: 89,836,302 |
| intron 1 CpG-34 | chr4: 89,836,281 |
| intron 1 CpG-35 | chr4: 89,836,273 |
| intron 1 CpG-36 | chr4: 89,836,262 |
| intron 1 CpG-37 | chr4: 89,836,260 |
| intron 1 CpG-38 | chr4: 89,836,248 |
| intron 1 CpG-39 | chr4: 89,836,243 |
| intron 1 CpG-40 | chr4: 89,836,241 |
| intron 1 CpG-41 | chr4: 89,836,231 |
| intron 1 CpG-42 | chr4: 89,836,228 |
| intron 1 CpG-43 | chr4: 89,836,201 |
| intron 1 CpG-44 | chr4: 89,836,129 |
| intron 1 CpG-45 | chr4: 89,836,102 |
| intron 1 CpG-46 | chr4: 89,835,989 |
| intron 1 CpG-47 | chr4: 89,835,904 |
| intron 1 CpG-48 | chr4: 89,835,730 |
| intron 2 CpG-1 | chr4: 89,835,383 |
| intron 2 CpG-2 | chr4: 89,835,353 |

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
