# Peer review of "SNCA Gene Methylation in Parkinson’s Disease and Multiple System Atrophy"

_2075-4655, 2022_

Round 1

Reviewer 1 Report

The manuscript of Fedotova et al. presents a methylation pattern of the MSA and PD in comparison to healthy controls in the SCNA gene. The SCNA gene showed hypomethylation of CpG sites in SNCA intron 1 in PD and hypermethylation of predominantly non-CpG sites in SNCA promoter region in MSA. In MSA patients hypermethylation in the promotor was associated with shorter disease duration.

The topic is important, but most of the studies are from brain tissue. The results are presented clearly. In the Introduction, I would highlight some issues that may not be clear to the broader audience. The methodology is sound.

Comments:

Are the age differences between the three groups (MSA, PD, controls) statistically significant?

I would develop on the following issue in the introduction: Line 207 Non-CpG methylation is relatively common in cells of the nervous system compared to other cells in the body. For readers not working on epigenetics it is not clear.

Are there any studies on how methylation pattern in blood are related to methylation in the brain?

I would make clear in the Discussion which studies are from blood and which are from the brain tissue (e.g. it is not clear for reference number 34).

Reviewer 2 Report

Review of a manuscript “SNCA gene methylation in Parkinson's disease and multiple system atrophy by E Fedotova and coauthors submitted to “Epigenome”

α-Synuclein is an important hallmark of Parkinson’s disease and other synucleinopathies. Its accumulation and aggregation play an important role in the pathogenesis of these diseases.  One of the reasons triggering its overproduction and accumulation is a defect in the regulation of α-synuclein expression due to the disbalance in the methylation of its gene. The authors examined hypomethylation of CpG sites in SNCA intron 1 in Parkinson’s disease patients and hypermethylation of non-CpG sites in promoter region of α-synuclein gene in multiple system atrophy patients. This is an important area of investigation and the results of the manuscript will be interesting for the readers of the journal.

The following corrections and additions should be made.

Abstract

Line 16: “In MSA patients hypermethylation in the promotor was associated with shorter disease duration”. The sense of this sentence is not clear, what is “shorter disease duration”?. The authors should rewrite it in a more understandable way.

Introduction

Lines 25-26: “…pathological aggregates of the small synaptic protein, alpha-synuclein, are found in the structures of the central and peripheral nervous system [2].” This is a clumsy sentence that can be corrected as follows :” pathological aggregates of the small synaptic protein, alpha-synuclein, are found in the central and peripheral nervous system [2].”

a loss of dopaminergic neurons n the substantia nigra pars compacta with the ac- 32

Line 32:”…accumulation of pathological aggregates of alpha-synuclein, Lewy bodies [8].”. This sentence should be rewritten as follows: ”accumulation of pathological aggregates of alpha-synuclein in Lewy bodies [8].”

Lines 34: ”PD is a multifactorial disease.” This is an awkward sentence which can be rewritten as follows: ”PD is a multifactorial disease with a documented substantial increase in its prevalence in the past decades; both environmental and genetic factors contribute to the pathology of this disorder [references : ”Biomarkers in Parkinson’s Disease”. Chapter in a book, editors Peplow PV et al. Neurodegenerative Diseases Biomarkers. 2022. Neuromethods, vol 173. pp 155-180. Humana, New York, NY. https://link.springer.com/protocol/10.1007/978-1-0716-1712-0_7.

Results

Line 132: “As one can be seen on Figures 1-4, in all the studied SNCA regulatory regions PD patients had an intermediate methylation level between MSA patients and control group.” This sentence should be rewritten as follows: ” As can be seen on Figures 1-4, PD patients had an intermediate methylation level between MSA patients and control group in all examined SNCA regulatory regions”

Line 135-136: ”In both synucleinopathies, PD and MSA, hypermethylation of SNCA intron 2 in the “body” of the gene) was determined.” The sentence should be corrected as follows: ”In both PD and MSA intron 2 was hypermethylated in the “body” of the SNCA gene.

Figure 2 The lower part of the Figure 2 below 40 does not contain information and can be truncated.

Figure 3 The upper part of the figure above 60 does not contain information and can be truncated. This will allow to better see the small figures and symbols in enlarged figure

Line 151:” We studied correlations between methylation levels and disease duration.” See our comments to the line 16 above.  

Line 165 “The presence of therapy with any antiparkinsonian drugs did not affect the methylation profile of the SNCA gene.” Should be rewritten as follows: ”The treatment  with any antiparkinsonian drugs did not affect the methylation profile of the SNCA gene.”

Line 166: ”Levodopa and therapy did not change the methylation pattern. The same was shown for amantadine therapy.” These two sentences should be combined as follows: ”Treatment by levodopa and amantadine did not change the methylation pattern.”

Discussion

Line 178: “Jowaed [25]” should be corrected as  “Jowaed et al. [25]”

Line 179: “Matsumoto [26]” should be corrected as “Matsumoto et al. [26]”

Materials and Methods

Line 253 :”We recruited 82 patients with a diagnosis of PD…” Was consent of patients or their relatives received?

Line 284: “Genomic DNA samples were obtained from peripheral blood leukocytes “ The reference on the method of DNA isolation should be given.

Line 286:” Analysis of the methylation of the SNCA gene was carried out by direct Sanger sequenching” reference on the method should be given.
